# Mortality and associated risk factors in patients hospitalized due to COVID-19 in a Peruvian reference hospital

Alonso Soto[1,2]*, Dante M. Quiñones-Laveriano[1], Johan Azañero[2,3],
Rafael Chumpitaz[2], José Claros[2], Lucia Salazar[2], Oscar Rosales[2], Liz Nuñez[4],
David Roca[4], Andres Alcantara[2]

**1** Instituto de Investigaciones en Ciencias Biomédicas (INICIB), Faculty of Medicine, Universidad Ricardo Palma, Lima, Peru, **2** Departament of Internal Medicine, Hospital Nacional Hipólito Unanue, Lima, Peru, **3** Universidad Científica del Sur, Lima, Peru, **4** Faculty of Medicine, Universidad Ricardo Palma, Lima, Peru

* alonso.soto@urp.edu.pe

**Data Availability Statement:** All relevant data are within the paper and its Supporting Information files.

## Abstract

### Objectives

To determine the risk factors for in-hospital mortality in patients with COVID-19 from a Peruvian national hospital.

### Methods

Retrospective cohort study of medical records of patients with COVID-19 hospitalized at Hospital Nacional Hipólito Unanue (HNHU) during the months of April to August 2020. The dependent variable was in-hospital mortality. Independent variables included sociodemographic and clinical characteristics, physical examination findings, oxygen saturation ($SaO_2$) at admission, treatment received during hospitalization and laboratory results at admission. A Cox regression model was used to evaluate the crude and adjusted hazard ratios for associated factors.

### Results

We included 1418 patients. Median age was 58 years (IQR 47–68 years) and 944 (66.6%) were male. The median length of hospitalization was 7 (4–13) days, and the mortality rate was 46%. The most frequent comorbidities were type 2 diabetes mellitus, hypertension, and obesity. In the adjusted analysis, mortality was associated with age (HR 1.02; 95%CI 1.02–1.03), history of surgery (HR 1.89; 95%CI 1.31–2.74), lower oxygen saturation at admission (HR 4.08; CI95% 2.72–8.05 for $SaO_2$<70% compared to $SaO_2$>94%), the presence of poor general condition (HR 1.81; 95% CI 1.29–2.53), altered state of consciousness (HR 1.58; 95%CI 1.18–2.11) and leukocyte levels (HR 1.01; 95%CI 1.00–1. 02). Treatment with ivermectin (HR 1.44; 95%CI 1.18–1.76) and azithromycin (HR 1.25; 95%CI 1.03–1.52) were associated with higher mortality. Treatment with corticosteroids at low to moderate doses was associated with lower mortality (HR 0.56 95%CI 0. 37–0. 86) in comparison to no steroid use.

**Funding:** The author(s) received no specific funding for this work.

**Competing interests:** The authors have declared that no competing interests exist.

## Conclusion

A high mortality was found in our cohort. Low oxygen saturation at admission, age, and the presence of hematological and biochemical alterations were associated with higher mortality. The use of hydroxychloroquine, ivermectin or azithromycin was not useful and was probably associated with unfavorable outcomes. The use of corticosteroids at moderate doses was associated with lower mortality.

## Introduction

COVID-19 pandemic is one of the most important sanitary catastrophes of our recent history and have had devastating consequences in health systems as well as quality of life of millions of people worldwide, especially in low and middle income countries [1], not only due to disease itself but also on its negative impact on healthcare workforce [2]. Until the end of December 2021, there were more than 278 million of cases and almost 5.45 million of deaths worldwide [3]. Peru, with more than 200,000 deaths due to COVID-19, is one of the countries with highest mortality [3, 4].

While most COVID-19 patients experience mild flu-like illness, a significant proportion develop pneumonia that can evolve into acute respiratory distress syndrome (ARDS), which requires hospitalization in critical care units [5, 6]. Within the group of hospitalized patients, mortality can be close to 50% [7], which can be even higher in older patients, those with comorbidities or with a lower oxygen saturation (SaO$_2$) at admission [5, 6]. Among laboratory markers, C-reactive protein (CRP), procalcitonin, lactate dehydrogenase (LDH) and platelets, among others, could predict a serious clinical deterioration in these patients [8–10]. However, predictors of mortality are heterogeneous among different countries. In addition, the association of several repurposed treatments for COVID with in-hospital outcomes needs to be addressed, particularly the most widespread drugs used during the first year of the pandemic, namely ivermectin, azithromycin and hydroxychloroquine

Knowledge of hospital epidemiology considering the different characteristics of population and health facilities is crucial for preparedness against COVID-19 pandemic. A better understanding of risk factors for mortality can facilitate the triage and management of patients and optimize the use of the limited resources available on public hospitals.

The objective of the study was to determine risk factors for in-hospital mortality in patients with COVID-19 from a Peruvian national hospital.

## Material and methods

### Study design and setting

The study was a retrospective cohort based on the review of clinical records of patients hospitalized due to COVID-19 hospitalized from April to August 2020 in Hospital Nacional Hipólito Unanue (HNHU), a reference hospital in Lima, Peru. HNHU coverage includes mainly uninsured population of poor socioeconomical background and is one of the hospitals that serves the largest number of COVID-19 patients and depends on Peruvian Ministry of Health (MINSA).

### Population

The population was composed of patients hospitalized in dedicated wards for care of patients with COVID-19. The diagnosis of COVID-19 was based on serological, molecular and/or

clinical radiological picture. Data from all patients hospitalized during the study period were taken, consisting of 1884 medical records. Statistical power was calculated to detect a HR considered clinically relevant of at least 1.5 with 95% confidence, resulting in a power greater than 99.9%. We included all records form hospitalized adult patients with diagnosis of COVID-19. We excluded patients with incomplete data regarding outcomes, and those who died or were discharged before 24 hours of admission.

### Data collection and variables

We retrospectively reviewed data from medical records using an *ad hoc* data collection form using anonymized data and then translated to a database in Microsoft excel. The dependent variable was in-hospital mortality. Independent variables included sociodemographic factors, clinical characteristics, comorbidities, vital functions at admission, oxygen saturation at admission, treatment received during hospitalization and laboratory data obtained at admission. Collection of data was coordinated with the archive of Hospital after authorization from Department of medicine and ethical clearance.

### Statistical analysis

Categorical variables were presented using frequencies and percentages, while numerical variables through measures of central tendency and dispersion. An initial bivariate analysis was performed using t-test for numerical variables and chi-squared for categorical ones. To evaluate the association of independent variable with mortality, Cox regression was performed to find the crude and adjusted hazard ratios (HR), with their respective 95% confidence intervals of those variables found significant in the bivariate analysis. The multivariate Cox regression model included those variables found significant in bivariate analysis except those variables considered as possibly collinear and those variables with less than 80% available values. All the analysis were preformed using STATA v15 for windows.

### Ethical aspects

The study was limited to the retrospective review of medical records. Collection of data were anonymized before entered in the database file. This study was conducted according to the tenets of the Declaration of Helsinki and was approved by the Department of Medicine of HNHU and by the Institutional Ethics Committees of HNHU (letter 116-2021-CIEI-HNHU) and Universidad Ricardo Palma (PG-25-2020).

## Results

Out of 1884 patients hospitalized in the study period, 1418 fulfill inclusion and exclusion criteria and were included in the final analysis (Fig 1).

The diagnosis was mainly serological (70.8%), followed by clinical radiological (21.5%) and molecular diagnosis with RT-PCR (7.7%). Median age was 58 years, and the most frequent gender was male (66.6%). The most frequent comorbidities were type 2 diabetes mellitus (DM), hypertension and obesity (Table 1).

The most common symptoms (Table 2) were dyspnea, cough, malaise, and headache while the most frequent findings on physical examination (Table 3) were tachypnea, tachycardia, and fever.

Most patients presented with leukocytosis, lymphopenia, high values of C reactive protein, and D-Dimer (Table 4).

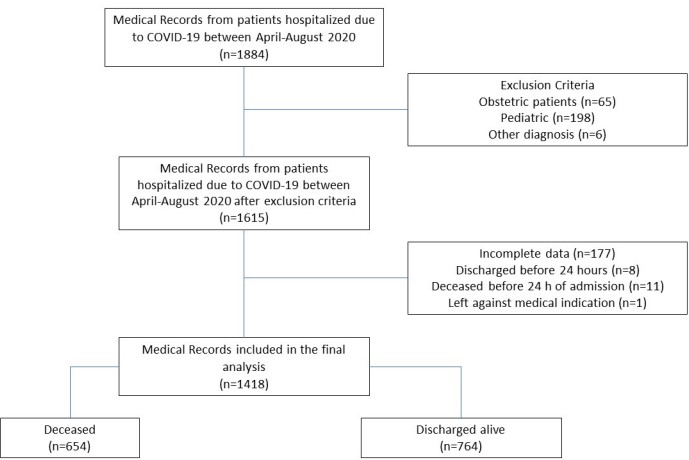

**Fig 1. Flow chart of patients included in the study.** Hospital Nacional Hipolito Unanue. April to August 2020. Lima, Peru.

Pharmacological treatments most frequently used (Table 5) were ivermectin (34%) hydroxychloroquine (41.6%), azithromycin (52.7%) and corticosteroids (88.1%). The median (interquartile range) of hospitalization days was 7 (4–13) days. The mortality rate was 46%.

## Bivariate analysis

Deceased patients were significantly older than survivors (64 vs 52 years; p<0.001). Among the antecedents, both diabetes mellitus (p = 0.03) and a history of surgery were associated with higher mortality (p<0.001). Duration of hospitalization was significantly shorter (p<0.001) among patients who eventually die (median 5 days; IQR 2–8) compared with those discharged alive (10 days; IQR 6–16 days). Among the symptoms, the presence of pharyngeal pain, diarrhea, nausea, and cough were associated with lower mortality, while the presence of myalgia, anorexia and dyspnea were associated with higher mortality. Regarding the findings on physical examination, an association of mortality was found with higher heart rate, respiratory rate and lower blood pressure and saturation at admission (Fig 2).

Mortality was also associated with evidence of cyanosis, intercostal retractions, and altered consciousness. Among the findings in the laboratory tests, the deceased patients presented significantly higher counts of leukocytes, and higher levels of C-reactive protein, D-dimer, alkaline phosphatase, lactate dehydrogenase, urea, creatinine and prothrombin time. On the other hand, their albumin level, lymphocyte, and platelet counts were significantly lower. With respect to pharmacological treatment, the use of ivermectin was associated with a mortality of 58% compared to 40% among those who did not receive it (p<0.001). Table 5 shows the pharmacological treatment received during hospitalization. Treatment with ivermectin, azithromycin, and high-dose systemic corticosteroids was found to be associated with higher mortality (Table 5 and Fig 3).

## Multivariate analysis

In the adjusted analysis (Table 6), mortality was associated with age (HR 1.02; 95%CI 1.02–1.03), history of surgery (HR 1.89; 95%CI 1.31–2.74), lower oxygen saturation at admission (HR 4.08; CI95% 2.72–8.05 for SatO2<70% compared to SatO2>94%), the presence of poor general condition (HR 1.81; 95% CI 1.29–2.53), altered state of consciousness (HR 1.58; 95% CI 1.18–2.11) and leukocyte levels (HR 1. 01; 95%CI 1.00–1. 02). Treatment with ivermectin

**Table 1. Sociodemographic characteristics, history, and its association with mortality in patients with COVID-19 admitted to the Hipólito Unanue National Hospital during April to August 2020.** Lima, Peru.

| Variables | Mortality | | Total (n = 1418)* | P-value |
|---|---|---|---|---|
| | Yes (n = 654) | No (n = 764) | | |
| Age | 64 (55–74) | 52 (42–62) | 58 (47–68) | <0.001 |
| Sex | | | | |
| Male | 449 (47.6%) | 495 (52.4%) | 944 (66.6%)* | 0.124 |
| Female | 205 (43.3%) | 269 (56.8%) | 474 (33.4%)* | - |
| Time from disease onset (n = 1008) | | | | |
| 5 days or less | 194 (54.0%) | 165 (46.0%) | 359 (35.6%)* | |
| 6 to 10 days | 233 (52.5%) | 211 (47.5%) | 444 (44.0%)* | 0.703 |
| 11 to 15 days | 78 (48.8%) | 82 (51.3%) | 160 (15.9%)* | |
| More than 15 days | 25 (55.6%) | 20 (44.4%) | 45 (4.5%)* | |
| Frailty | | | | |
| Yes | 42 (87.5%) | 6 (12.5%) | 48 (3.4%)* | <0.001 |
| No | 612 (44.7%) | 758 (55.3%) | 1370 (96.6%)* | - |
| Cancer | | | | |
| Yes | 10 (58.8%) | 7 (41.2%) | 17 (1.2%)* | 0.291 |
| No | 644 (46.0%) | 757 (54.0%) | 1401 (98.8%)* | - |
| Surgery | | | | |
| Yes | 45 (66.2%) | 23 (33.8%) | 68 (4.8%)* | 0.001 |
| No | 609 (45.1%) | 741 (54.9%) | 1350 (95.2%)* | - |
| Tuberculosis | | | | |
| Yes | 33 (46.5%) | 38 (53.5%) | 71 (5.1%)* | 0.951 |
| No | 621 (46.1%) | 726 (53.9%) | 1347 (94.9%)* | - |
| Asthma | | | | |
| Yes | 18 (43.9%) | 23 (56.1%) | 41 (2.9%)* | 0.772 |
| No | 636 (46.2%) | 741 (53.8%) | 1377 (97.1%)* | - |
| Diabetes mellitus | | | | |
| Yes | 126 (40.7%) | 184 (59.3%) | 310 (21.9%)* | 0.029 |
| No | 528 (47.7%) | 580 (52.3%) | 1108 (78.1%)* | - |
| High blood pressure | | | | |
| Yes | 149 (49.8%) | 150 (50.2%) | 299 (21.1%)* | 0.147 |
| No | 505 (45.1%) | 614 (54.9%) | 1119 (78.9%)* | - |
| Obesity | | | | |
| Yes | 133 (50%) | 133 (50%) | 266 (18.8%)* | 0.159 |
| No | 521 (45.2%) | 631 (54.8%) | 1152 (81.2%)* | - |

*Percentage over all patients.

(HR 1.44; 95%CI 1.18–1.76) and azithromycin (HR 1.25; 95%CI 1.03–1.52) were associated with higher mortality. On contrary, treatment with corticosteroids at low to moderate doses was associated with lower mortality (HR 0.56; 95%CI 0. 37–0. 86).

## Discussion

Our cohort showed a high mortality rate (46%) like other Peruvian hospital cohorts [5–8, 10, 11] which reported a mortality between 33% and 50%. Peru is the third country in Latin America with the highest mortality beneath Brazil and Mexico [12]. In contrast to Europe and Asia, Latin America [13] has high COVID-19 death rates. Probably one of the reasons for the high

**Table 2. Symptoms reported in patients hospitalized for COVID 19 admitted to Hospital Nacional Hipólito Unanue from April to August 2020.** Lima, Peru.

| | Mortality | | | |
|---|---|---|---|---|
| *Respiratory symptoms* | Yes (n = 654) | No (n = 764) | Total (n = 1418)* | P-value |
| Sore throat /odynophagia (n = 1416) | | | | |
| Yes | 64 (37.2%) | 108 (62.8%) | 172 (12.1%)* | 0.012 |
| No | 590 (47.4%) | 654 (52.6%) | 1244 (87.9%)* | - |
| Rhinorrhea (n = 1416) | | | | |
| Yes | 15 (36.6%) | 26 (63.4%) | 41 (2.9%)* | 0.211 |
| No | 639 (46.5%) | 736 (53.5%) | 1391 (97.1%)* | - |
| Dyspnea/shortness of breath (n = 1416) | | | | |
| Yes | 546 (48.8%) | 574 (51.2%) | 1120 (79.1%)* | <0.001 |
| No | 108 (36.5%) | 188 (63.5%) | 296 (20.9%)* | - |
| Chest/back pain (n = 1416) | | | | |
| Yes | 69 (44.8%) | 85 (55.2%) | 154 (10.9%)* | 0.716 |
| No | 585 (46.3%) | 677 (53.7%) | 1262 (89.1%)* | - |
| Cough (n = 1416) | | | | |
| Yes | 333 (43.4%) | 434 (56.6%) | 767 (54.2%)* | 0.023 |
| No | 321 (49.5%) | 328 (50.5%) | 649 (45.8%)* | - |
| *Gastrointestinal Symptoms* | | | | |
| Diarrhea (n = 1416) | | | | |
| Yes | 19 (29.7%) | 45 (70.3%) | 64 (4.5%)* | 0.007 |
| No | 635 (47.0%) | 717 (53.0%) | 1352 (95.5%)* | - |
| Abdominal pain (n = 1416) | | | | |
| Yes | 23 (27.1%) | 62 (72.9%) | 85 (6.1%)* | <0.001 |
| No | 631 (47.41%) | 700 (52.6%) | 1331 (93.9%)* | - |
| Nausea/vomiting (n = 1416) | | | | |
| Yes | 29 (31.9%) | 62 (68.1%) | 91 (6.4%)* | 0.005 |
| No | 625 (47.1%) | 700 (52.9%) | 1325 (93.6%)* | - |
| **General Symptoms** | | | | |
| Headache (n = 1416) | | | | |
| Yes | 71 (39.4%) | 109 (60.6%) | 180 (12.7%)* | 0.052 |
| No | 583 (47.2%) | 653 (52.8%) | 1236 (87.3%)* | - |
| Myalgias/arthralgias (n = 1416) | | | | |
| Yes | 9 (20.0%) | 36 (80.0%) | 38 (2.7%)* | <0.001 |
| No | 645 (47.1%) | 726 (52.9%) | 1378 (97.3%)* | - |
| Hiporexia/anorexia (n = 1416) | | | | |
| Yes | 30 (60.0%) | 20 (40.0%) | 50 (3.5%)* | 0.046 |
| No | 624 (45.7%) | 742 (54.3%) | 1366 (96.5%)* | - |
| General malaise (n = 1416) | | | | |
| Yes | 190 (46.7%) | 217 (53.3%) | 407 (28.7%)* | 0.812 |
| No | 464 (46.0%) | 545 (54.0%) | 1009 (71.3%)* | - |
| Fatigue/asthenia (n = 1416) | | | | |
| Yes | 14 (33.3%) | 28 (66.7%) | 42 (2.9%)* | 0.090 |
| No | 640 (46.6%) | 734 (53.4%) | 1374 (97.1%)* | - |
| Diaphoresis (n = 1416) | | | | |
| Yes | 6 (46.1%) | 7 (53.9%) | 13 (0.9%)* | 0.998 |
| No | 648 (46.2%) | 755 (53.8%) | 1403 (99.1%)* | - |

*Percentage over total cases.

**Table 3. Clinical characteristics and mortality risk in patients with COVID-19 admitted to the Hospital Nacional Hipólito Unanue from April to August 2020.** Lima, Peru.

| Variables | Mortality | | Total (n = 1418) | P-value |
|---|---|---|---|---|
| | Yes (n = 654) | No (n = 764) | | |
| *Vital signs* | | | | |
| Systolic blood pressure (n = 642) | 110 (100–120) | 110 (100–124) | 110 (100–120) | 0.018 |
| Diastolic blood pressure (n = 640) | 70 (60–70) | 70 (60–80) | 70 (60–78) | 0.059 |
| Heart rate (n = 1339) | 101 (90–113) | 99 (86–111) | 100 (88–112) | 0.002 |
| Respiratory rate (n = 949) | 28 (23–30) | 24 (20–28) | 24 (22–30) | <0.001 |
| Oxygen saturation at admission (n = 1323) | | | | <0.001 |
| ≥95% | 51 (21.4%) | 187 (78.6%) | 238 (17.9%)* | |
| 90%– 94% | 64 (22.9%) | 215 (77.1%) | 279 (21.2%)* | |
| 85%– 89% | 76 (29.8%) | 179 (70.2%) | 255 (19.3%)* | |
| 80%– 84% | 100 (55.9%) | 79 (44.1%) | 179 (13.5%)* | |
| 75%– 79% | 84 (74.3%) | 29 (25.7%) | 113 (8.5%)* | |
| 70%– 74% | 55 (73.3%) | 20 (26.7%) | 75 (5.7%)* | |
| <70% | 164 (89.1%) | 20 (10.9%) | 184 (13.9%)* | |
| *Physical Examination* | | | | |
| Cyanosis (n = 1416) | | | | |
| Yes | 14 (93.3%) | 1 (6.7%) | 15 (1.1%)* | <0.001 |
| No | 640 (45.7%) | 761 (54.3%) | 1401 (98.9%)* | - |
| Fever (n = 1418) | Yes | No | | |
| Yes | 224 (44.4%) | 280 (55.6%) | 504 (35.5%)* | 0.347 |
| No | 430 (47.1%) | 484 (52.9%) | 914 (64. 5%)* | - |
| Pallor (n = 1416) | | | | |
| Yes | 19 (59.4%) | 13 (40.6%) | 32 (2.3%)* | 0.130 |
| No | 635 (45.9%) | 749 (54.1%) | 1384 (97.7%)* | - |
| Retractions (n = 1416) | | | | |
| Yes | 23 (71.9%) | 9 (28.1%) | 32 (2.3%)* | 0.003 |
| No | 631 (45.6%) | 753 (54.4%) | 1384 (97.7%)* | - |
| Poor general condition (n = 1416) | | | | |
| Yes | 81 (84.4%) | 15 (15.6%) | 96 (6.8%)* | <0.001 |
| No | 573 (43.4%) | 747 (56.6%) | 1320 (93.2%)* | - |
| Alteration of consciousness (n = 1416) | | | | |
| Yes | 100 (74.1%) | 35 (25.9%) | 135 (9.5%)* | <0.001 |
| No | 554 (43.2%) | 727 (56.8%) | 1281 (90.5%)* | - |

*Percentage over total.

mortality rate is the socioeconomic and demographic inequality that Latin America faces [14] leading to inequitable access to poorly prepared health services with extremely limited access to critical care. In fact, only 4% of patients in our cohort were hospitalized in intensive care services despite the large percentage of patients with requirement of being admitted to these units. Another factor that could explain the high mortality was hypoxemia on admission, in accordance with what was described by Bahl et al. [15], Dienderé et al. [16] and Mejia et al. [7]. Most of the hospitalized patients had decreased arterial oxygen saturation levels, with an increase in mortality proportional to the degree of hypoxemia. This reflects a late admission which can be explained due to lack of access to hospital beds. Another associated factor was advanced age, also shown in other studies [17]. Advanced age is associated with decline in

**Table 4. Laboratory findings in patients with COVID-19 admitted to the Hospital Nacional Hipólito Unanue from April to August 2020.** Lima, Peru.

| | Mortality | | | |
|---|---|---|---|---|
| Variables (at admission) | Yes (n = 654) | No (n = 764) | Total (n = 1418) | *P-value* |
| leukocytes x $10^3$/μL (n = 1304) | 14 (10.4–18) | 10.4 (7.8–14.2) | 11.7 (8.7–16) | <0.001 |
| lymphocytes x $10^3$/μL (n = 1303) | 0.7 (0.5–1.1) | 1 (0.6–1.4) | 0.9 (0.6–1.3) | <0.001 |
| Neutrophils x $10^3$/μL (n = 1080) | 12.3 (8.8–16.6) | 8.65 (6.2–12.4) | 10.4 (7.2–14.8) | <0.001 |
| Platelets x $10^3$/μL (n = 1310) | 268 (205–351) | 308.5 (239–390) | 288.5 (225–378) | <0.001 |
| Hemoglobin gr/dL (n = 1310) | 13.3 (12.2–14.5) | 13.2 (12.2–14.3) | 13.3 (12.2–14.4) | 0.199 |
| PCR (n = 1091) | 18.1 (10.2–25.1) | 12.4 (4.8–19.9) | 15.0 (6.6–22.3) | <0.001 |
| D-Dimer (n = 926) | 2.1 (1.0–5.1) | 1.0 (0.6–2.0) | 1.4 (0.7–3.8) | <0.001 |
| Fibrinogen (n = 989) | 729 (582–889) | 727.5 (598–875.5) | 728 (593–881) | 0.822 |
| Ferritin (n = 196) | 1012.4 (595.9–1722.2) | 865 (544.5–1329) | 938.2 (574.1–1493.7) | 0.119 |
| Lactate dehydrogenase (n = 856) | 461.9 (371.9–612.1) | 313.6 (246.7–415.1) | 385.9 (284.6–509.1) | <0.001 |
| Aspartate aminotransferase (n = 989) | 45.9 (30.4–67.7) | 40.7 (24.6–68) | 43 (27.4–68) | 0.007 |
| Alanine aminotransferase (n = 988) | 46.8 (29.1–80.1) | 55.1 (29.1–93.8) | 49.6 (29.1–88.5) | 0.023 |
| Alkaline phosphatase (n = 973) | 127.1 (96.1–176.2) | 113.6 (84.9–160.3) | 119.1 (90–168.5) | <0.001 |
| Gamma Glutamyl Transpeptidase (n = 965) | 93.1 (44.3–184) | 101 (49.4–195.1) | 95.4 (47.3–192.6) | 0.162 |
| Albumin (n = 969) | 3.1 (2.8–3.4) | 3.4 (3.1–3.7) | 3.2 (2.9–3.6) | <0.001 |
| Prothrombin time (seconds) (n = 996) | 15.5 (14.3–16.9) | 14.6 (13.7–15.6) | 15 (14–16.3) | <0.001 |
| International Normalized Ratio (n = 990) | 1.2 (1.1–1.3) | 1.1 (1.1–1.2) | 1.1 (1.1–1.2) | <0.001 |
| Glucose (n = 999) | 136.7 (109.5–182.6) | 128.9 (103.9–179.3) | 132.8 (106.1–181.2) | 0.105 |
| Urea (n = 1162) | 43.0 (31.5–65.1) | 30.8 (23–42.8) | 35.6 (25.5–52.4) | <0.001 |
| Creatinine (n = 1178) | 0.74 (0.6–1.01) | 0.67 (0.55–0.81) | 0.69 (0.57–0.87) | <0.001 |

**Table 5. Treatment in patients with COVID-19 admitted to the Hospital Nacional Hipólito Unanue from April to August 2020.** Lima, Peru.

| | Mortality | | | |
|---|---|---|---|---|
| Variable | Yes (n = 654) | No (n = 764) | Total (n = 1418) | *P-value* |
| **Ivermectin** | | | | |
| Yes | 280 (57.9%) | 204 (42.2%) | 484 (34.2%)* | <0.001 |
| No | 374 (40.0%) | 560 (60.0%) | 934 (65.8%)* | - |
| **Hydroxychloroquine** | | | | |
| Yes | 292 (49.5%) | 298 (50.5%) | 590 (41.6%)* | 0.032 |
| No | 362 (43.7%) | 466 (56.3%) | 828 (58.4%)* | - |
| **Azithromycin** | | | | |
| Yes | 352 (52.7%) | 316 (47.31%) | 668 (47.1%)* | <0.001 |
| No | 302 (40.3%) | 448 (59.7%) | 750 (52.9%)* | - |
| **Systemic corticosteroids**** (n = 1417)** | | | | |
| No corticosteroids | 125 (28.9%) | 308 (71.1%) | 433 (30.5%)* | - |
| < 50mg/day | 50 (29.8%) | 118 (70.2%) | 168 (11.9%)* | <0.001 |
| ≥50mg/day | 479 (58.7%) | 337 (41.3%) | 816 (57.6%)* | |
| **Admission to the ICU (n = 1190)** | | | | |
| Yes | 31 (67.4%) | 15 (32.6%) | 46 (3.9%)* | 0.084 |
| No | 623 (54.5%) | 521 (45.5%) | 1144 (96.1%)* | - |

*Percentage according to total observations

**methylprednisolone dose equivalent.

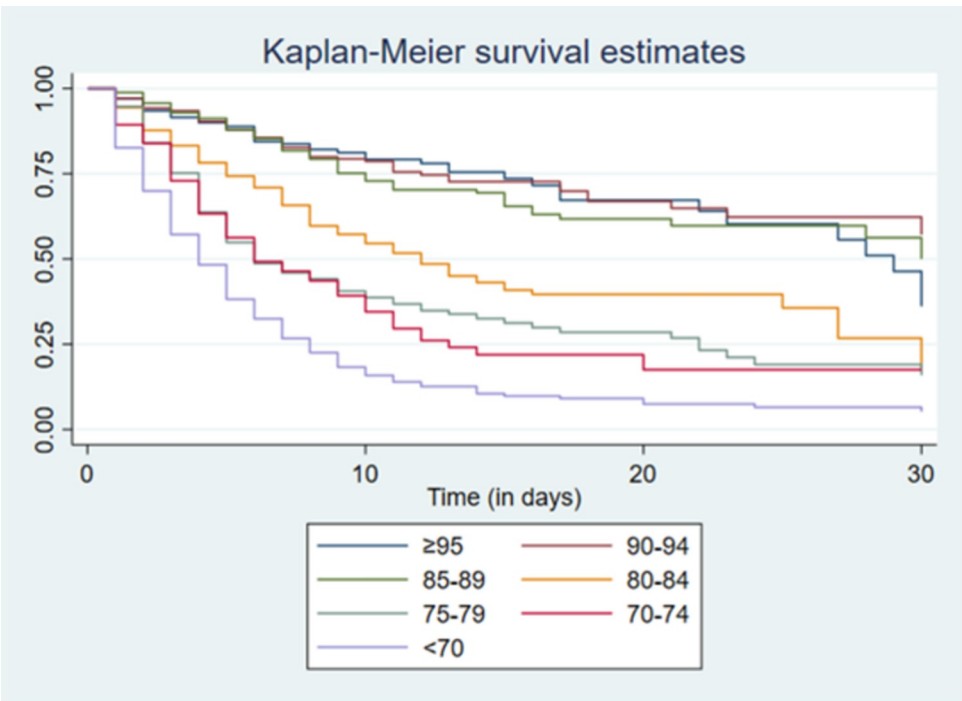

**Fig 2. Kaplan-Meier survival curves comparing oxygen saturation of hemoglobin (SaO2) at admission in patients hospitalized with COVID-19 between April and August 2020 in Hospital Nacional Hipolito Unanue.** Lima, Peru. Log rank test: p<0,001.

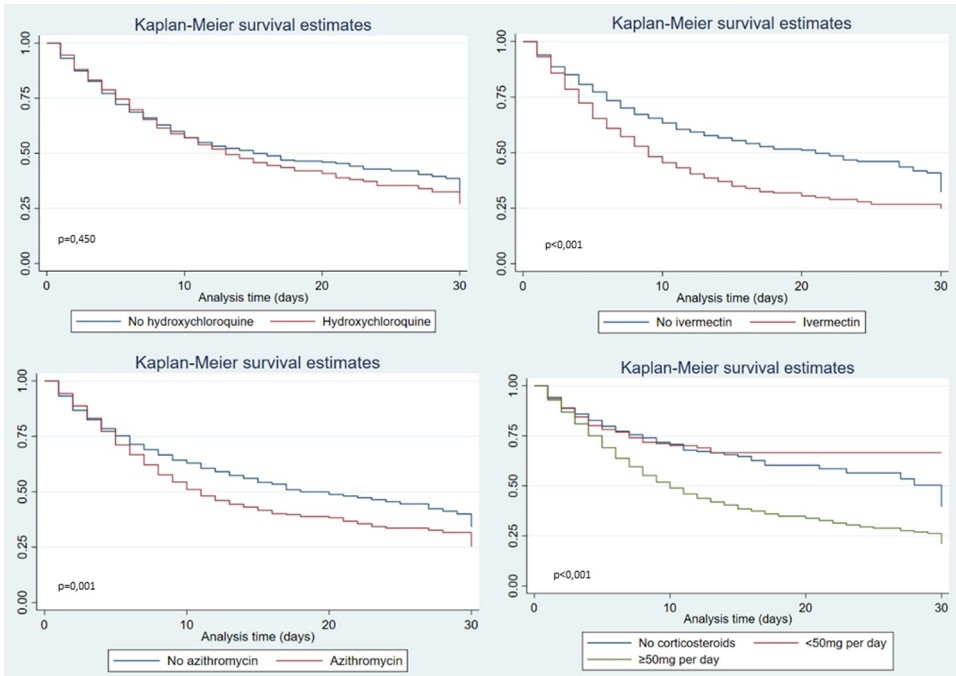

**Fig 3. Kaplan-Meier survival curves comparing treatment used in in patients hospitalized with COVID-19 between April and August 2020 in Hospital Nacional Hipolito Unanue.** Lima, Peru. P values shown are based on log rank-test.

**Table 6. Mortality risk factors in COVID-19 patients hospitalized at Hospital Nacional Hipólito Unanue from April to August 2020.** Results from multivariate analysis.

| Variable | HRc (IC95%) | p value | HRa (IC95%) | p value |
|---|---|---|---|---|
| Age | 1.04 (1.03–1.04) | <0.001 | 1.02 (1.02–1.03) | <0.001 |
| Sore throat/odynophagia (n = 1416) | 0.68 (0.53–0.88) | 0.004 | 0.90 (0.65–1.25) | 0.531 |
| Myalgias/arthralgias (n = 1416) | 0.39 (0.20–0.75) | 0.005 | 0.39 (0.16–0.96) | 0.040 |
| Cough (n = 1416) | 0.78 (0.67–0.91) | 0.002 | 0.87 (0.71–1.06) | 0.166 |
| Heart rate (n = 1339) | 1.01 (1.00–1.01) | 0.001 | 1.00 (0.99–1.01) | 0.701 |
| Oxygen Saturation | | | | |
| ≥95% | Ref. | - | Ref. | - |
| 90%– 94% | 0.94 (0.65–1.35) | 0.727 | 0.86 (0.55–1.34) | 0.497 |
| 85%– 89% | 1.09 (0.77–1.56) | 0.623 | 1.05 (0.68–1.61) | 0.833 |
| 80%– 84% | 2.22 (1.59–3.12) | <0.001 | 1.96 (1.29–2.97) | 0.002 |
| 75%– 79% | 3.26 (2.31–4.63) | <0.001 | 2.78 (1.82–4.25) | <0.001 |
| 70%– 74% | 3.72 (2.54–5.45) | <0.001 | 3.31 (2.06–5.32) | <0.001 |
| <70% | 5.86 (4.28–8.03) | <0.001 | 4.07 (2.72–6.09) | <0.001 |
| *History and Physical Examination* | | | | |
| Cyanosis (n = 1416) | 3.95 (2.32–6.72) | <0.001 | 1.98 (0.86–4.58) | 0.109 |
| Fever (n = 1418) | 0.89 (0.75–1.04) | 0.145 | 1.07 (0.87–1.32) | 0.514 |
| Poor general condition (n = 1416) | 3.17 (2.51–4.00) | <0.001 | 1.80 (1.29–2.51) | 0.001 |
| Alteration of consciousness (n = 1416) | 2.51 (2.03–3.11) | <0.001 | 1.58 (1.18–2.11) | 0.002 |
| *Symptoms* | | | | |
| Sore throat/odynophagia (n = 1416) | 0.68 (0.53–0.88) | 0.004 | 0.90 (0.65–1.25) | 0.531 |
| Myalgias/arthralgias (n = 1416) | 0.39 (0.20–0.75) | 0.005 | 0.39 (0.16–0.96) | 0.040 |
| Cough (n = 1416) | 0.78 (0.67–0.91) | 0.002 | 0.87 (0.71–1.06) | 1.166 |
| *Laboratory findings* | | | | |
| Leucocytes x$10^3$ (n = 1304) | 1.01 (1.01–1.01) | <0.001 | 1.01 (1.00–1.02) | 0.010 |
| Lymphocites x$10^3$ (n = 1303) | 0.80 (0.70–0.91) | <0.001 | 1.01 (0.88–1.15) | 0.892 |
| Platelets (n = 1310) | 0.99 (0.99–0.99) | <0.001 | 0.99 (0.99–0.99) | <0.001 |
| Urea per 10mg/dL (n = 1162) | 1.04 (1.02–1.05) | <0.001 | 1.03 (1.00–1.05) | 0.004 |
| *Medical History* | | | | |
| Previous Surgery (n = 1418) | 1.74 (1.29–2.36) | <0.001 | 1.86 (1.29–2.70) | 0.001 |
| Diabetes mellitus (n = 1418) | 0.87 (0.71–1.05) | 0.146 | Not included | - |
| *Drug treatment* | | | | |
| Ivermectin (n = 1418) | 1.60 (1.36–1.85) | <0.001 | 1.41 (1.16–1.72) | <0.001 |
| Azithromycin (n = 1418) | 1.30 (1.12–1.52) | 0.001 | 1.25 (1.03–1.53) | 0.021 |
| Hydroxychloroquine (n = 1418) | 1.06 (0.91–1.23) | 0.463 | Not included | - |
| Systemic Corticosteroids (n = 1417)* | | | | |
| No corticosteroids | Ref. | - | Ref. | - |
| Less than 50mg/day | 0.87 (0.62–1.20) | 0.386 | 0.56 (0.36–0.86) | 0.008 |
| ≥ 50mg/day | 1.85 (1.52–2.26) | <0.001 | 1.04 (0.78–1.39) | 0.776 |

HRc: Crude Hazard Ratio HRa: adjusted Hazard Ratio

*methylprednisolone dose equivalent.

both humoral and cellular immune responses [18], and if fragility is added, we will have even more ominous figures. In fact, we found a mortality rate in frail older adults of 88%, a figure consisting with other reports [19–21].

Comorbidities such as diabetes, hypertension and obesity were not significantly associated with mortality in both bivariate and multivariate analyses, in contrast to what was reported by

Corona et al. [22] Escobedo-de la Peña et al. [23] and Poly et al. [24]. Although it could be hypothesized that there were undiagnosed patients with those comorbidities at admission, both blood pressure and glycemia at admission did not show association with mortality. Peruvian studies found contradictory results regarding these factors as predictors of mortality in patients hospitalized for COVID-19 [5, 7, 25]. Latin American studies found mixed results. A Mexican study found that DM and obesity, but not hypertension, were associated with mortality in hospitalized patients [26]; while a nationwide study in Brazil found an association with diabetes and hypertension, but not with obesity [27] and a Chilean study showed that obesity, but not diabetes mellitus was an independent risk factor for mortality [28].

One hypothesis that could explain these results is that these conditions may be associated with the risk of being hospitalized in the general population, but that they do not necessarily have an association with mortality in patients with already severe COVID-19, which was the condition of the majority of our patients. A study in around 4000 critical patients in Italy found no association with hypertension [29], and other studies in critical patients found no association between mortality and diabetes [30, 31] or obesity [32, 33]. Some authors suggest that this could be due to the "obesity paradox" or "diabetes paradox", hypotheses that suggest that these conditions are not a risk factor for mortality in already critically ill patients, as has been widely seen in other diseases [34, 35]. However, these hypotheses are still controversial and are not entirely clear in the COVID-19 context.

Due to resource constraints, we were not able to evaluate cardiovascular markers like troponins, nor had electrocardiographic and echocardiographic data which could have allowed us to properly assess cardiovascular disease, which has been correlated with an increased risk of COVID-19 infection and poorer prognosis [36, 37], increasing mortality risk up to three times according to two recent metanalysis [38, 39].

Hematological and biochemical alterations in COVID-19 [40] have proven useful in the stratification of COVID-19 patients, and many of these have served as predictors of mortality. Regarding hematological findings, mortality was associate with leukocytosis and lower numbers of lymphocytes and platelets, as well as significantly higher levels of D-dimer and lactate dehydrogenase. Those findings are consistent with previous studies [36, 37].

Although several clinical trials are underway, and the benefit of some drugs like moderate doses of corticosteroids and tocilizumab in patients with severe cases, no drug can cure or prevent COVID-19. One striking finding of our cohort was the association between the pharmacological treatments provided and mortality. The association between ivermectin use and mortality is of concern. Although treatment with this drug has been advocated by some members of the medical community, and even included in Peruvian national guidelines, there is no evidence of its usefulness. Moreover, a tendency to higher mortality was found with the use of this drug [41]. As for hydroxychloroquine–an antimalarial currently used in lupus erythematosus and rheumatoid arthritis–widely used during the first wave of the pandemic, our findings suggest not only its ineffectiveness but its possible deleterious effect. The Solidarity [42] clinical trial also showed a trend towards higher mortality. Although caution was suggested against the indiscriminate use of this drugs [43], its use continued for several months. A previous study conducted in hospitals of the Social Health Security of Peru–EsSalud, showed that the use of ivermectin, azithromycin and hydroxychloroquine were not effective against COVID-19, and were even associated with higher mortality [41]. One of the few drugs that has shown evidence in decreasing mortality in hypoxemic patients is dexamethasone [44, 45]. Our results show a consistent effect in cases of low to moderate dose corticosteroids, while higher doses were associated with higher mortality in the bivariate analysis (although not in the adjusted analysis). The use of high doses should therefore be assessed only in the context of clinical trials. These findings should lead us to reflect on the use of drugs without evidence, which unfortunately

has been one of the interventions promoted even as a national policy against COVID-19 [46]. The use of interventions without scientific support, even in health emergency situations such as the one experienced, should not be repeated.

Our study presents as limitations the difficulty in obtaining a diagnosis by molecular methods in most patients. Most patients presented a diagnosis based on a typical radiological clinical picture and/or a positive serology. However, in the context of the huge epidemic faced and the very high incidence of COVID-19 during the study period, it is unlikely that an appreciable proportion of patients with other pathologies were included in the cohort. Another limitation is the retrieval of clinical data on acutely and severely ill patients. Moreover, physical examination is often limited in the context of the biosecurity measures necessary for the care of these patients and diagnostic aids are not always available. However, working conditions in a pandemic context are far from ideal and we consider that our data not only represent a valuable contribution to understanding the hospital epidemiology of COVID-19, but also allow us to analyze the association of clinical variables and treatments and their impact with mortality during the so-called first wave of COVID-19 in Peru.

## Conclusion

Patients hospitalized for COVID-19 during the first wave in a Peruvian national reference hospital presented a high mortality. A low $SaO_2$ at admission, age, and the presence of hematological and biochemical alterations were associated with higher mortality. The use of drugs with purported antiviral activity such as hydroxychloroquine, ivermectin or azithromycin was not useful and is probably associated with unfavorable outcomes. The use of corticosteroids at moderate doses was associated with lower mortality. However, high doses were not associated with a better prognosis and are probably associated with higher mortality.

## Supporting information

**S1 Data.**
(DTA)

## Author Contributions

**Conceptualization:** Alonso Soto, Dante M. Quiñones-Laveriano, Johan Azañero, Rafael Chumpitaz, José Claros, Lucia Salazar, Andres Alcantara.

**Data curation:** Alonso Soto, Dante M. Quiñones-Laveriano, Rafael Chumpitaz, José Claros, Lucia Salazar, Oscar Rosales, Liz Nuñez, David Roca, Andres Alcantara.

**Formal analysis:** Alonso Soto, Dante M. Quiñones-Laveriano, Johan Azañero.

**Investigation:** Alonso Soto, Dante M. Quiñones-Laveriano, Johan Azañero, Rafael Chumpitaz, José Claros, Lucia Salazar, Oscar Rosales, Liz Nuñez, David Roca, Andres Alcantara.

**Methodology:** Alonso Soto, Dante M. Quiñones-Laveriano.

**Software:** Alonso Soto.

**Supervision:** Alonso Soto.

**Validation:** Alonso Soto.

**Visualization:** Alonso Soto.

**Writing – original draft:** Alonso Soto, Dante M. Quiñones-Laveriano.

**Writing – review & editing:** Alonso Soto, Dante M. Quiñones-Laveriano, Johan Azañero, Rafael Chumpitaz, José Claros, Lucia Salazar, Oscar Rosales, Liz Nuñez, David Roca, Andres Alcantara.

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
