## [Decision Letter · Decision Letter 0]

14 Dec 2021

PONE-D-21-33659Mortality and associated risk factors in patients hospitalized due to COVID-19 in a Peruvian reference hospital.PLOS ONE

Dear Dr. Soto,

Thank you for submitting your manuscript to PLOS ONE. After careful consideration, we feel that it has merit but does not fully meet PLOS ONE’s publication criteria as it currently stands. Therefore, we invite you to submit a revised version of the manuscript that addresses the points raised during the review process.

The manuscript is interesting but some improvements are needed.

We look forward to receiving your revised manuscript.

Kind regards,

Prof. Raffaele Serra, M.D., Ph.D

Academic Editor

PLOS ONE

Journal Requirements:

Reviewers' comments:

Reviewer's Responses to Questions

**Comments to the Author**

1. Is the manuscript technically sound, and do the data support the conclusions?

Reviewer #1: Yes

2. Has the statistical analysis been performed appropriately and rigorously? 

Reviewer #1: Yes

3. Have the authors made all data underlying the findings in their manuscript fully available?

Reviewer #1: Yes

4. Is the manuscript presented in an intelligible fashion and written in standard English?

Reviewer #1: No

5. Review Comments to the Author

Reviewer #1: The authors aimed to determine risk factors for in-hospital mortality in patients with COVID-19 from a Peruvian national hospital.

The findings are interesting but there are no valid comments on cardiovascular disease (except hypertension) that are important factors for mortality in COVID-19 patients. I think in the discussion section the importance of cardiovascular disease must be deepened. For example cite and discuss the paper by Ielapi N. et al. Cardiovascular disease as a biomarker for an increased risk of COVID-19 infection and related poor prognosis. Biomark Med. 2020 Jun;14(9):713-716.

Overall considered discussion section is too superficial and must be better tailored on the main aspects of the paper.

6. PLOS authors have the option to publish the peer review history of their article (what does this mean?). If published, this will include your full peer review and any attached files.

Reviewer #1: No

---

## [Author Response · Author response to Decision Letter 0]

24 Jan 2022

Dear Editor:

Following the reviewers and editor evaluation of the original research article entitled “Mortality and associated risk factors in patients hospitalized due to COVID-19 in a Peruvian reference hospital” we have included the suggestions in the manuscript, which is sent as separates files with and without track changes. 

Below, you will find the answers to the specific comments. 

1. Please ensure that your manuscript meets PLOS ONE's style requirements, including 

those for file naming. 

Answer

We have updated the article formatting to fulfill PLOS ONE´s style requirements

2. Please provide additional details regarding participant consent. In the ethics statement in the Methods and online submission information, please ensure that you have specified (1) whether consent was informed and (2) what type you obtained (for instance, written or verbal, and if verbal, how it was documented and witnessed). If your study included minors, state whether you obtained consent from parents or guardians. If the need for consent was waived by the ethics committee, please include this information. If you are reporting a retrospective study of medical records or archived samples, please ensure that you have discussed whether all data were fully anonymized before you accessed them and/or whether the IRB or ethics committee waived the requirement for informed consent. If patients provided informed written consent to have data from their medical records used in research, please include this information.

Answer

The study was limited to the retrospective review of medical records. Collection of data were anonymized before entered in the database file. The study was approved by the Department of Medicine of HNHU and by the Institutional Ethics Committees of HNHU (letter 116-2021-CIEI-HNHU) and Universidad Ricardo Palma (PG-25-2020). The scanned version of both letters are included in the new submission.

Answer

Due to a misunderstanding, we believe that the standard procedure was giving the data upon request. However, we are willing to share the database of the study. Therefore, we have upload the anonymized dataset.

Reviewer Comments to the Author

Reviewer #1: The authors aimed to determine risk factors for in-hospital mortality in patients with COVID-19 from a Peruvian national hospital.

The findings are interesting but there are no valid comments on cardiovascular disease (except hypertension) that are important factors for mortality in COVID-19 patients. I think in the discussion section the importance of cardiovascular disease must be deepened. For example cite and discuss the paper by Ielapi N. et al. Cardiovascular disease as a biomarker for an increased risk of COVID-19 infection and related poor prognosis. Biomark Med. 2020 Jun;14(9):713-716.

Overall considered discussion section is too superficial and must be better tailored on the main aspects of the paper.

Answer

We have updated and improved the discussion, expanding the scope including relevant studies regarding cardiovascular and metabolic diseases. We added several additional references including the one suggested by the reviewer. 

While revising your submission, please upload your figure files to the Preflight Analysis and Conversion Engine (PACE) digital diagnostic tool, https://pacev2.apexcovantage.com/. PACE helps ensure that figures meet PLOS requirements. 

Answer 

We have updated the figures using the PACE tool. They are now compatible with PLOS requirements.

---

## [Editor Report · Decision Letter 1]

17 Feb 2022

Mortality and associated risk factors in patients hospitalized due to COVID-19 in a Peruvian reference hospital.

PONE-D-21-33659R1

Dear Dr. Soto,

We’re pleased to inform you that your manuscript has been judged scientifically suitable for publication and will be formally accepted for publication once it meets all outstanding technical requirements.

Kind regards,

Prof. Raffaele Serra, M.D., Ph.D

Academic Editor

PLOS ONE

Additional Editor Comments (optional):

amended manuscript is acceptable
---

## [Editor Report · Acceptance letter]

22 Feb 2022

PONE-D-21-33659R1 

Mortality and associated risk factors in patients hospitalized due to COVID-19 in a Peruvian reference hospital. 

Dear Dr. Soto:

I'm pleased to inform you that your manuscript has been deemed suitable for publication in PLOS ONE. Congratulations! Your manuscript is now with our production department. 

Kind regards, 

on behalf of

Prof. Raffaele Serra 

Academic Editor

PLOS ONE